# Detection of lymphoproliferative disease virus in Iowa Wild Turkeys (*Meleagris gallopavo*): Comparison of two sections of the proviral genome

**Kelsey C. Smith** *[ORCID]* *[*][◎], **Julie A. Blanchong**[◎]

Department of Natural Resource Ecology and Management, Iowa State University, Ames, Iowa, United States of Ameria

◎ These authors contributed equally to this work.
* smith2kc@gmail.com

**Data Availability Statement:** All relevant data are within the manuscript and its Supporting Information files.

## Abstract

An accurate diagnostic test is an essential aspect of successfully monitoring and managing wildlife diseases. Lymphoproliferative Disease Virus (LPDV) is an avian retrovirus that was first identified in domestic turkeys in Europe and was first reported in a Wild Turkey (*Meleagris gallopavo*) in the United States in 2009. It has since been found to be widely distributed throughout North America. The majority of studies have utilized bone marrow and PCR primers targeting a 413-nucleotide sequence of the *gag* gene of the provirus to detect infection. While prior studies have evaluated the viability of other tissues for LPDV detection (whole blood, spleen, liver, cloacal swabs) none to date have studied differences in detection rates when utilizing different genomic regions of the provirus. This study examined the effectiveness of another section of the provirus, a 335-nucleotide sequence starting in the U3 region of the *LTR* (Long Terminal Repeat) and extending into the Matrix of the *gag* region (henceforth *LTR*), for detecting LPDV. Bone marrow samples from hunter-harvested Wild Turkeys (n = 925) were tested for LPDV with the *gag* gene and a subset (n = 417) including both those testing positive and those where LPDV was not detected was re-tested with *LTR*. The positive percent agreement (PPA) was 97.1% (68 of 70 *gag* positive samples tested positive with *LTR*) while the negative percent agreement (NPA) was only 68.0% (236 of 347 *gag* negative samples tested negative with *LTR*). Cohen's Kappa (κ = 0.402, Z = 10.26, p<0.0001) and the McNemar test (OR = 55.5, p<0.0001) indicated weak agreement between the two gene regions. We found that in Iowa Wild Turkeys use of the *LTR* region identified LPDV in many samples in which we failed to detect LPDV using the *gag* region and that *LTR* may be more appropriate for LPDV surveillance and monitoring. However, neither region of the provirus resulted in perfect detection and additional work is necessary to determine if *LTR* is more reliable in other geographic regions where LPDV occurs.

**Funding:** Funding for this work was provided by Iowa State University (https://www.iastate.edu/) to author JAB. This paper is a product of the Iowa Agriculture and Home Economics Experiment Station, Ames, Iowa. Project No.'s IOWA5434 and IOWA5615 are sponsored by Hatch Act and State of Iowa funds. The funders had no role in study design, data collection and analysis, decision to publish, or preparation of the manuscript.

**Competing interests:** The authors have declared that no competing interests exist.

## Introduction

Wildlife disease surveillance is important for multiple reasons: disease can negatively impact biodiversity, human health, domestic animal and livestock health, and economic stability [1]. Surveillance can aid not only with mitigation, but also with preventing the spread of disease to new areas. To successfully monitor and manage wildlife diseases, it is essential to have an accurate method of detection, otherwise there is a risk of inaccurately estimating factors such as prevalence, virulence, and transmission [2]. Imperfect diagnostic tests can also hinder our ability to predict spread [2] and result in the allocation of resources to inefficient management plans [3].

Lymphoproliferative disease virus (LPDV) is an avian retrovirus first identified during the 1970's in Europe and Israel, and again in North America in 2009 [4]. While its entire geographic distribution has yet to be determined, to date it has been identified in Wild Turkeys (*Meleagris gallopavo*) along the East coast [5–7], as far West as Colorado [4, 8], as far south as Texas [9] and in several Canadian provinces: Manitoba, Ontario, and Quebec [10, 11]. Although previous outbreaks in Europe and Israel were in domestic turkeys, in the United States, natural infection has only been identified in Wild Turkeys [4]. Compared to their domestic counterparts, that experienced heightened levels of internal lesions and mortality [12], prior North American studies utilizing hunter-harvested turkeys have indicated that infection in wild birds is typically asymptomatic [4, 8, 10, 11, 13], although mortalities have been observed [4]. Clinical symptoms, when present, are typically non-specific, such as lethargy or anorexia [12]. To date, little is known of the transmission mechanism [4], population-level effects, and impacts on poults (chicks) [11].

Retroviruses function by integrating their RNA genome into host DNA, so as DNA replicates, the virus does as well [14, 15]. Viral RNA is transcribed into DNA via reverse transcriptase and is then integrated into the host's genome. Organization of the LPDV provirus is as follows: 5'-*LTR-gag-pro-pol-env-LTR*-3' [4, 16]. The complete proviral genome of the first LPDV case in North America (12/AR/2009) was 7,432 nucleotides long including *LTR*s [4]. Virus expression and replication first occur in bone marrow, then distribute to other lymphoid tissues [8, 16]. In a prior study by Thomas et al. [8] it was found that bone marrow was the most efficient tissue for diagnosing LPDV in turkeys obtained from hunter-harvest, compared to spleen and liver.

Diagnosis of LPDV is typically through polymerase chain reaction (PCR) utilizing DNA obtained from bone marrow, although whole blood and cloacal swabs have also been found to be a suitable alternative in live birds [13, 17]. When testing for LPDV, the partial p31/ partial capsid region of the *gag* gene is widely used among researchers [4, 8, 10, 11, 13]. However, this region may not consistently detect infection due to variability among viral strains across the distribution of turkeys (Renshaw R., Cornell University, personal communication). An exploratory examination of LPDV detection in samples collected from turkeys in both the eastern and western distribution of their range suggested that a different section of the LPDV provirus beginning in the U3 region of the Long Terminal Repeat and extending into the *gag* Matrix (henceforth referred to as *LTR*) may be more accurate at detecting infected birds (Renshaw R., Cornell University, personal communication). Long terminal repeats are an essential element of retroviruses, containing components essential for integration and transcription of the provirus [18], gene expression, and insertional mutagenesis [19]. Studies of other retroviruses have also found that PCR primers targeting different regions of the proviral genome had varying detection rates [20, 21].

With Wild Turkey populations suspected to be declining (as indicated by observation of declines in spring harvest numbers) across the United States due to unknown causes [22],

some natural resource managers are interested in expanding our understanding of LPDV and its potential role in declines. To do so, it is important to use the best available diagnostic test. The goal of this study was to determine if there are differences in LPDV detection rates when testing the same set of samples with the *gag* and *LTR* sections of the provirus and whether using *LTR* leads to the detection of more LPDV positive Wild Turkeys in Iowa than are detected using *gag*.

## Methods

### Sample collection

Hunter-harvested Wild Turkey tarsi were collected between 2019 and 2021 (n = 1022) in cooperation with the Iowa Department of Natural Resources (IADNR). The county and location of harvest was provided for each bird. Samples were also provided by the IADNR State Wildlife Veterinarian (submitted for necropsy) and from road-killed turkeys. Upon receipt, tarsi were stored in a -20˚C or -80˚C freezer until processing. Because samples were collected from hunter-harvested or road-killed animals the Iowa State University Intuitional Animal Care and Use Committee (IACUC) did not require an IACUC protocol for this study.

### Bone marrow and DNA extraction

A 10% bleach solution followed by water and ethanol was used to clean and flame sterilize equipment prior to each use to prevent cross-contamination when removing bone marrow from tarsi. Legs were cleaved, then forceps and probes used to extract bone marrow. Approximately 25mg was aliquoted into a 1.5mL microcentrifuge tube for DNA extraction; samples were refrigerated until extraction. Remaining tissue was stored in a separate tube and frozen at -20˚C or -80˚C.

The DNeasy Blood & Tissue Kit (Qiagen) was used to extract DNA from bone marrow using the manufacturer's protocol, with the following modification: elution of DNA was performed with 100μL of AE buffer instead of 200μL to increase final DNA concentration. If a small quantity of bone marrow was extracted (i.e., approximately half the amount of bone marrow typically used or less), samples were instead eluted with 50μL. A Denovix DS-11 spectrophotometer (NanoDrop Technologies) was used to quantify the amount of DNA extracted. A working sample for PCR was then created by diluting samples with sterile water to a standard concentration of 50ng/μL to ensure consistency across samples. Some samples (approximately 40%) had too small an amount of bone marrow to create the standard concentration working sample and were left undiluted (concentration of undiluted samples ranged from 8.01 ng/μL to 49.85ng/μL).

### PCR

PCR using primers developed by Allison et al. [4] amplifying the p31 and a part of the capsid domains within the virus' *gag* gene was conducted to test samples for LPDV. PCR was conducted following the protocol described in Alger et al. [13] with slight modifications to the PCR mix and reaction conditions, as follows. Products were amplified with a Mastercycler EP Gradient (Eppendorf) in a 20μL reaction consisting of 10μL HotStarTaq Plus Master Mix (Qiagen), 1μL of each 10μM primer (forward: 5'-ATGAGGACTTGTTAGATTGGTTAC-3', reverse: 5'-TGATGGCGTCAGGGCTATTTG-3' [4]), 6μL water, and 2μL DNA (concentration ranging from 8.01–50.0ng/μL following standardization). Reaction conditions included a denaturation step at 95˚C for 5 min, followed by denaturation at 95˚C for 30 sec, annealing at 54˚C for 30 sec, and extension at 72˚C for 1 min for 34 cycles, ending with extension at 72˚C for 10 min.

A negative and positive control was included in each PCR reaction, where the negative control contained water rather than DNA template. Positive controls were spleen/liver tissues from New York turkeys that were confirmed LPDV positive through DNA sequencing. Initial positive controls were of limited quantity, so a subset of samples that tested positive were purified with the QIAquick PCR Purification Kit (Qiagen) to isolate DNA for use as additional positive controls; of these, four were sequenced for confirmation of LPDV infection. Agarose gel electrophoresis was used to visualize PCR products, where the presence of a 413 bp band matching that of the positive control was used to determine if a sample was positive for LPDV infection (Gel 1 in S1 File). Infection was confirmed through sequencing using the forward primer for a subset (50%) of positive samples.

To compare the detection rates between the two sections of the provirus (*gag* and *LTR*), 45% of samples that tested *gag* positive and 45% of samples that tested *gag* negative were randomly selected and tested with *LTR*. The amplification protocol for testing with *LTR* was similar to that of the *gag* gene but the annealing temperature was increased to 56˚C to reduce nonspecific binding. Primers designed to amplify a large portion of the 5' *LTR* (starting in the U3 region) and extending into the Matrix domain of the *gag* gene were used (forward: 5'–GGGCACGGGATTGGCTT-3'; reverse:– 5'–AAACGCTCAATACACGACACAAC–3', Renshaw R., Cornell University, personal communication). Results were visualized through agarose gel electrophoresis following the same protocol as with the *gag* gene (Gel 2 in S1 File). A subset of samples that tested positive with *LTR* (some of which were *gag* positive and some of which were *gag* negative) were sequenced with the forward *LTR* primer for confirmation of LPDV infection (6%).

## Sensitivity and specificity analysis

The total number of samples that tested *gag+/LTR+*, *gag+/LTR-*, *gag-/LTR-*, and *gag-/LTR+* was determined to compare the detection rate of LPDV with the *LTR* segment of the provirus against that of the *gag* gene (reference test). In order to examine where results from the *gag* and *LTR* regions were in agreement (i.e., a sample is diagnosed as positive by both methods or negative by both methods), the Positive Percent Agreement (PPA) and Negative Percent Agreement (NPA) were calculated. Although the standard among researchers, results from this study (as described below) indicated that detection of LPDV with the *gag* gene was imperfect. The use of imperfect reference tests during the calculation of sensitivity and specificity introduces bias and may result in the underestimation of true values [23]. In cases such as this, an alternative is to calculate the PPA and NPA [24, 25]. Rather than indicating how often a novel test (*LTR* in this study) is correct or incorrect, PPA and NPA describe how often the novel test is in agreement or disagreement with the reference test. Calculations are performed in the same manner as sensitivity and specificity, where PPA is the portion of *gag* positive samples that also tested positive with *LTR* out of the total number of *gag* positives and NPA is the portion of *gag* negative samples that also tested negative with *LTR* out of the total number of *gag* negatives.

## Statistical analysis

While useful in determining where results from *gag* and *LTR* were in agreement, PPA and NPA are not statistical assessments. To statistically compare results from *gag* and *LTR*, Cohen's kappa and the McNemar test were used. Cohen's kappa was used to statistically examine how often the results were in agreement (i.e., samples tested positive with both *gag* and *LTR* or negative with both *gag* and *LTR*). Cohen's kappa is a statistical test measuring the agreement between two different raters or tests that describes the overall agreement between the two tests

while accounting for agreement due to chance [26]. Cohen's kappa tests the null hypothesis that any agreement is due to chance ($\kappa = 0$) against the alternative hypothesis that agreement is not a result of chance ($\kappa \neq 0$) [27]. The 'Kappa' function from the R package 'vcd' was used for computations [28, 29] and was run as unweighted because there were two nominal variables. Cohen's kappa value was interpreted according to the scale suggested by McHugh: 0–0.20, none; 0.21–0.39, minimal; 0.40–0.59, weak; 0.60–0.79, moderate; 0.80–0.90, strong; above 0.90, almost perfect agreement [26].

Another method of comparing results from *gag* and *LTR* is with the McNemar test, which tests agreement between proportions of the discordant cells in a contingency table [30]. This test looks at where diagnostic tests disagree rather than overall agreement using the null hypothesis that both types of disagreement are equally likely against the alternative that they are not equally likely. The McNemar test was used to examine how often diagnosis with *LTR* differed from *gag* and if those differences were proportional (i.e., whether there were equal or unequal numbers of *gag+/LTR-* and *gag-/LTR+*). Of the four versions of the McNemar test (classical, continuity corrected, exact, and mid-P), evidence suggests that mid-P provides the best combination of the higher power seen in classical tests and the reduced type I error rate seen in corrected tests [30, 31]. The function 'exact2x2' from the R package 'exact2x2' was used to perform the McNemar test [29, 32, 33]. Data were treated as paired and the mid-p value was used. The McNemar odds ratio (OR) obtained from analysis is calculated by dividing the number of *gag-/LTR+* by the number of *gag+/LTR-* and represents how much more likely a result of *gag-/LTR+* is than *gag+/LTR-* [34]. The OR may be interpreted according to the following: $< 1$, less likely; $= 1$, equally likely; $> 1$, more likely. Significance levels for both Cohen's Kappa and the McNemar test were set at $\alpha \leq 0.05$. All analyses were performed in R (version 4.1.1).

## Results

Extracted DNA ranged in concentration from 8.01ng/μL to 1275.26ng/μL prior to standardization. A total of 925 samples were of sufficient quality for testing and had unambiguous results with the *gag* gene. Of these, LPDV was undetected in 772 samples and 153 tested positive using the *gag* gene (S1 Data). Infection status appeared to be unrelated to DNA concentration, with samples of both low and high DNA concentrations testing LPDV positive using both *gag* and *LTR*. A subset (n = 417)– 45% of *gag* positive and 45% of *gag* negative samples–was randomly selected for paired testing with *LTR* (Table 1). The subset of samples that tested positive with *LTR* that were sequenced with *LTR* primers all most closely matched LPDV sequences in the National Center for Biotechnology Information (NCBI) BLAST database (pairwise identity and percent identical ranged from 93.4% to 99.0%), confirming detection of LPDV with *LTR*. *Gag* and *LTR* sequences were deposited in GenBank under the accession numbers OR026188-OR026260 and OR026261-OR026271, respectively.

The positive percent agreement (PPA) was 97.1% (68/70) and the negative percent agreement (NPA) was 68.0% (236/347), indicating greater disagreement with *LTR* for samples testing *gag* negative than samples testing *gag* positive (Table 2). Cohen's kappa ($\kappa = 0.402$) was significant (Z = 10.26, p < 0.0001), falling between the categories of minimal and weak

**Table 1. Pairwise comparison of LPDV detected (+) and undetected (-) samples using the *gag* and LTR sections of the provirus.**

|         | GAG (+) | GAG (-) | Total |
|---------|---------|---------|-------|
| LTR (+) | 68      | 111     | 179   |
| LTR (-) | 2       | 236     | 238   |
| Total   | 70      | 347     | 417   |

**Table 2. Evaluation of LTR as a diagnostic test for LPDV infection against the standard (*gag*) using positive percent agreement (PPA), negative percent agreement (NPA), Cohen's kappa, and McNemar's odds ratio (OR).**

|  | *RESULT* | *95% CI* | *P-VALUE* |
|---|---|---|---|
| *PPA* | 97.1% |  |  |
| *NPA* | 68.0% |  |  |
| *COHEN'S KAPPA* | 0.402 | 0.325–0.479 | <0.0001 |
| *MCNEMAR OR* | 55.5 | 16.5–334.5 | <0.0001 |

agreement between the results from *gag* and *LTR* (Table 2). The McNemar mid-p test was also significant (OR = 55.5, p < 0.0001), indicating the number of *gag* positive/*LTR* negative and *gag* negative/*LTR* positive samples was disproportionate, and that there were significantly more samples that tested positive with *LTR* and negative with *gag*. (Table 2).

## Discussion

This study was performed to examine differences in detection rates when testing Iowa Wild Turkeys for LPDV with the *gag* and *LTR* sections of the provirus, therefore assessing the utility of *LTR* for detection of LPDV. The data demonstrated that there was high agreement in results between *gag* and *LTR* in samples that were *gag* positive, with both testing positive in 97.1% of *gag* positive samples. However, there was a high level of disagreement for samples that were *gag* negative, with both testing negative in only 68.0% of *gag* negative samples. *LTR* resulted in the identification of an additional 111 samples as LPDV positive. Sequencing confirmed that *LTR* was detecting the presence of LPDV in turkeys. These results show that, at least in Wild Turkeys in Iowa, the *LTR* section of the LPDV provirus is able to detect infected individuals including those testing negative with the *gag* gene, and that testing with the *gag* gene results in false negatives. Although *LTR* did detect more positive animals than *gag* in this study, it did not detect all cases. One of the two *gag* positive but *LTR* undetected cases was sequenced with *gag* primers to confirm LPDV infection and that the disagreement was not due to lab error.

The high mutation rate in viruses can lead to large amounts of genetic variation [35, 36], which is exhibited in LPDV by the many different strains identified across studies based on the *gag* gene [4, 8, 10, 11, 37–39]. PCR is used to detect LPDV infections by using primers that bind to a specific region of the provirus that has been inserted into the turkey genome, but may fail to detect the virus even when it is present when there are high levels of variation that lead to mismatches between primers and the binding region of the sample [35]. Because of this, diagnostic methods that are successful in one geographic region may be less successful in another geographic region. As a virus spreads among individuals, mutations will occur, creating increasing amounts of diversity between nucleotide sequences as the infection spreads. The results of this study suggest that the *LTR* section may be a more conservative region of the LPDV provirus and thus less likely to contain genetic variation that would negatively affect primer binding success in comparison to the *gag* gene (Renshaw R., Cornell University, personal communication).

This study included only samples from Iowa Wild Turkeys, which represents a small part of the Wild Turkey range and the distribution of LPDV that has been documented to date [4–11]. Because of the geographic limitation of this study, it is not yet clear if *LTR* performs similarly well at detecting LPDV in Wild Turkeys in other geographic regions. Another limitation is that a relatively small number of *gag* positive samples were tested with *LTR*. A larger sample size might provide a more accurate estimate of *LTR*'s false negative rate. Further, while sequenced samples consistently matched LPDV sequences in the NCBI BLAST database and

re-testing samples reinforced confidence in positive results, not all *gag* or *LTR* tested samples were sequenced, thus there is the possibility of false positives.

Future research could include testing for LPDV with *LTR* in additional regions covering the rest of the Wild Turkey's range to determine if *LTR* performs similarly in other geographic regions, or if its ability to detect additional positive turkeys is restricted to the Midwest. Testing could also be conducted in subspecies other than Eastern Wild Turkeys. Other sections of the LPDV provirus or whole genome sequencing could also be considered for viability as a diagnostic test, as could other types of PCR depending on sample type (e.g., real-time, reverse-transcriptase). While nucleotide conservation was previously found to be similar across the four viral genes of LPDV (*gag*, *pro*, *pol*, *env*), *pol* was found to have the highest level of conservation (90.1%), compared to *gag* (88.3%), *pro* (88.3%), and *env* (86.6%) [4]. This might suggest *pol* as another viable option for testing. Further, a recent study of LPDV in Wild Turkeys in Texas employed quantitative PCR (qPCR) and the *env* gene for surveillance [9]. Universal primers have been developed for other viruses and could provide another possible option for future research [35].

Successfully monitoring and managing LPDV in Wild Turkey populations requires an accurate method of detection [2, 3]. Detection methods with a high rate of false negatives might result in an underestimation of prevalence and incorrectly identifying risk factors or areas of concern [2]. If *LTR* performs similarly in other geographic regions, it would suggest that LPDV prevalence might be higher than has previously been reported. By correctly identifying positive cases, additional risk factors may come to light, or previously identified variables may prove to be less significant than initially believed, allowing for better predictions. Identifying risk factors can help identify areas potentially at risk and identifying areas with high numbers of cases might suggest areas to target for future studies on LPDV. This in turn can lead to more efficient management and a better allocation of resources [3].

## Supporting information

**S1 Data. CSV file containing the Wild Turkey LPDV infection status dataset.**
(CSV)

**S1 File.**
(DOCX)

## Acknowledgments

We would like to thank Randall Renshaw from Cornell University who shared his hypothesis about differences in LPDV detectability using *gag* vs *LTR* that led us to conduct this study and for providing samples for use as positive controls. We thank the Iowa Department of Natural Resources staff and hunters for their assistance in obtaining samples from harvested Wild Turkeys.

## Author Contributions

**Conceptualization:** Kelsey C. Smith, Julie A. Blanchong.

**Data curation:** Kelsey C. Smith.

**Formal analysis:** Kelsey C. Smith.

**Funding acquisition:** Julie A. Blanchong.

**Investigation:** Kelsey C. Smith.

**Methodology:** Kelsey C. Smith, Julie A. Blanchong.

**Resources:** Julie A. Blanchong.

**Supervision:** Julie A. Blanchong.

**Validation:** Kelsey C. Smith.

**Visualization:** Kelsey C. Smith.

**Writing – original draft:** Kelsey C. Smith, Julie A. Blanchong.

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
