## [Decision Letter · Decision Letter 0]

16 Oct 2023

PONE-D-23-30258Detection of lymphoproliferative disease virus in Iowa Wild Turkeys (Meleagris gallopavo): Comparison of two sections of the viral genomePLOS ONE

Dear Dr. Smith,

Thank you for submitting your manuscript to PLOS ONE. After careful consideration, we feel that it has merit but does not fully meet PLOS ONE’s publication criteria as it currently stands. Therefore, we invite you to submit a revised version of the manuscript that addresses the points raised during the review process.

Improving diagnostic test specificity and sensibility to provide a most accurate picture of an epidemiologic scenario is critical for planning adaguate measures to curb an
infectious disease spread and minimizing its impacts on several sectors. To this end, new approaches are constantly needed to keep up with disease dynamics that evolve in conjunction with changing geographic range, transition between different host organisms, and new pathogen
variants.

This study provides solid and reliable results to pave the way to a more effective diagnostic assay to detect LPDV among wild turkeys in North America. Below points need to be addressed.

1. Abstract

Line 30: I would like to recommend removing "the portion" of the *gag* gene (or if you prefer to mention it, it would be better to specify the sequence) and replacing "virus' genome" with provirus if it's what the authors
were referring to. Please consider following this suggestion in other sentences of the paper.

Line 31: .......other tissues.... Could you specify?

Line 32:..... different sections....Could you articolate better?

Line 33: Could the authors please rephrase the purpose? I suggest changing the following part " of another section of the viral genome a portion of the" at most.

Line 69: I think the proviral genome definition is not necessary. 

Line 80: The authors should remove (*LTRs)*

Line 115: "Some sample...". Could you please clarify? A few samples? Do these poor-quality samples can lead to biased estimations?

2. Methods.

Line 122: I suggest to explain what "slight modifications" are.

Line 122-124: the authors should indicate the concentration instead of volume.

Line 130: "Positive controls ......were confirmed LPDV positive." Could the authors mention the methodology? I would remove "undefined", which does not add significant experimental value.

Line136: Could you please mention the gene the forward primer is complementary to?

Line 142-143: Please, indicate the 3’ and 5’ ends of primers.

Line 149: The title of this section seems very general to me. You could divide the content into two separate sections: sensitivity and sensibility analysis, and statistical analysis.

3. Results:

I suggest the authors expand the paragraph emphasizing the robustness of the results.

Line 199: "Infection status ...testing positive". Could you please explain which gene the samples are positive for?

4. Discussion

Line 238: I suggest removing "it was not a perfect test" and leaving after the comma "it did not detect all cases".

As a general suggestion, I recommend adding more references whenever the authors mention literature data. I also suggest the authors avoid general terms and redundant definitions whenever possible.

We look forward to receiving your revised manuscript.

Kind regards,

Elisabetta Pilotti

Academic Editor

PLOS ONE

Journal Requirements:

"We would like to thank Randall Renshaw from Cornell University who shared his hypothesis about differences in LPDV detectability using gag vs LTR that led us to conduct this study and for providing samples for use as positive controls. We thank the Iowa Department of Natural Resources staff and hunters for their assistance in obtaining samples from harvested Wild Turkeys. Funding for this work was provided by Iowa State University. This paper is a product of the Iowa Agriculture and Home Economics Experiment Station, Ames, Iowa. Project No.’s IOWA5434 and IOWA5615 are sponsored by Hatch Act and State of Iowa funds."

"Iowa State University (https://www.iastate.edu/) provided funding to author JAB. This paper is a product of the Iowa Agriculture and Home Economics Experiment Station, Ames, Iowa. Project No.’s IOWA5434 and IOWA5615 are sponsored by Hatch Act and State of Iowa funds. The funders had no role in study design, data collection and analysis, decision to publish, or preparation of the manuscript."

Reviewers' comments:

Reviewer's Responses to Questions

**Comments to the Author**

1. Is the manuscript technically sound, and do the data support the conclusions?

Reviewer #1: Yes

Reviewer #2: Yes

2. Has the statistical analysis been performed appropriately and rigorously? 

Reviewer #1: Yes

Reviewer #2: Yes

3. Have the authors made all data underlying the findings in their manuscript fully available?

Reviewer #1: Yes

Reviewer #2: Yes

4. Is the manuscript presented in an intelligible fashion and written in standard English?

Reviewer #1: Yes

Reviewer #2: Yes

5. Review Comments to the Author

Reviewer #1: The study evaluated whether two PCR targets (p31/CA and 5’ LTR/MA) were more reliable for detecting lymphoproliferative disease virus (LPDV) from Eastern wild turkey bone marrow samples than one PCR target alone (p31/CA). The authors determined a high positive percent agreement between the two targets; however, the negative percent agreement was not similar. Statistical analysis did demonstrate a weak agreement between the two targets. The results of this study are of value in improving the diagnostic tests for LPDV, an emerging avian retrovirus.

The manuscript is well-written, with a clear rationale, methodology, results, and conclusions, and limitations of the study were discussed. The sample size is large, with nearly 1,000 samples tested. The two statistical tests used (Cohen’s kappa test and McNemar test) were appropriate. The tables and table captions are clear and readable. In addition, the positive controls were extracted DNA from previously diagnosed LPDV-positive wild turkey samples.

The negative control, though, was water in place of DNA. Preferably, the negative control would be extracted DNA from a known uninfected wild turkey. However, the authors have results from extracted DNA from wild turkeys that they determined were LPDV-negative, which can be used as a comparison. In addition, a control targeting a wild turkey gene would have been useful for showing that the DNA extractions did not contain any PCR inhibitors. However, the study primarily focuses on whether one LPDV target is amplified when the other LPDV target from the same sample is. Further, the samples were tested only once, and only a subset of amplicons were sequenced to verify that the PCR products contained the targeted sequences. Overall, though, the study and conclusions are likely reproducible.

DNA sequences were deposited into GenBank, and the manuscript contains the accession numbers. In addition, IACUC protocols were not required since they received hunter-harvested animals and conformed to ethical guidelines.

Major Revision Suggestions: none.

Minor Revision Suggestions:

1. Line 197: has written that 924 samples were tested. However, on Line 201: 772 samples tested negative, and 153 tested positive, which adds up to 925 samples. Please confirm the sample size tested and results and clarify as needed.

2. Readers would benefit from seeing representative gels containing the resolved PCR products (positive and negative controls; positive and negative samples) as Supporting Information.

Reviewer #2: Overall: This is a very interesting and thorough study. It is certainly enlightening that the LTR region was able to detect positives that were not detected by the standard gag gene. This is definitely useful to the field going forward. Nice work and well-written. I do think you could go into more detail in both the intro and the discussion about the viral genes, why the LTR region may be a more conserved region compared with the gag gene. Has this been shown in other viruses? I think further justification and explanation on this topic of the comparison of genes and viruses in general is required for the reader to appreciate the full interpretation and impact of this work. Lastly, I might mention somewhere that you are detecting pro-viral DNA (you explain the incorporation of retroviruses into the genome well, just important to be clear that you are detecting the inserted DNA rather than the original RNA). Also, general comment, since you only use 924 samples, this should be your “n” rather than the 1,022 you started with.

Abstract:

Line 29: Could be worth also mentioning that blood and other tissues (organs, cloacal swabs) have also been used.

Line 31: Then here could mention that the viability of the various sample types mentioned above have been evaluated, but not the portion of the viral genome (enhancing your justification!).

Line 33: Use commas to separate the “a portion of the LTR” clause.

Intro:

Line 63: If being inclusive, also Shea et al. 2022 in JWD - https://doi.org/10.7589/JWD-D-21-00152 - 699 asymptomatic turkeys

Line 64: Could be useful to mention some nonspecific symptoms, and that Allison et al. 2014 did report that it can cause mortality just to further justify your work

Paragraph Lines 75-84: I think stronger justification is needed as to how the genes may differ in their detection ability. I’m not sure personal communication is a strong enough justification. Has it been shown that detection varies between genes in other viruses? I’m not disagreeing, but I do think the justification and evidence for the justification is slightly lacking here. The last line is informative and does help with the justification, but since your whole premise is assessing another gene, providing sound justification for this is important.

Line 85: It seems like you are referencing a diagnostic paper to indicate that wild turkey populations are declining. The reports of decline are somewhat isolated, and I would be more specific here and cite the sources that identified these declines.

Methods:

Line 106: Since you indicate below in line 112 that some samples had less initial material, it would be helpful to indicate the range of bone marrow mg used. Or you can add this into line 112 and put it in parenthesis, for instance: “If a small quantity of bone marrow was extracted (10-24mg), samples were instead eluted with 50ul (to increase yield).” Since you are specifically looking at detection probability, these somewhat specific details are important.

Lines 115-117: Again, useful to include the range of those samples. Were you potentially working with 1ng/ul? This may be a discussion point too if samples that had lower concentrations varied in their detection rates among genes.

Lines 122-124: for primers and DNA, it is useful to use concentrations. You’ve already touched upon DNA concentrations, so at 50ng/ul you are using 100ng, but you did indicate this varies, so could include the range (though still note you attempted to standardize because this is good to know). Along these lines, it could be helpful to know how many were standardized, was it 90% of your samples? 10% of your samples? Providing these details help give the bigger picture of the work. For primers, it could be helpful to put the final concentration in parenthesis, or that you used 1ul of each 10uM primer.. etc.

Line 135: Just curious why the 386 bp instead of the 413 typically seen with the gag gene and these primers?

Lines 127-137: Could probably save some text here by indicating your negative control was water instead of DNA template, and then I’m not sure you have to indicate where you got the positive control, but just that it was confirmed positive via sequencing – you indicated confirmed positive, I assume that’s by sequencing? I might just add that tidbit.

Line 137: Could include the amount that you confirmed through sequencing (% of positive samples confirmed via sequencing).

Line 140: Instead of processing, could be more specific and say the thermocycle protocol.

Lines 142-145: Depending on the journal it may be necessary to just indicate the 3’ and 5’ ends of primers. Also, did Renshaw develop these primers? I’m not sure if the journal may want some information as to the development of these primers at all.

Line 146: Same comment as above in regards to providing the amount in the subset.

Analysis:

Line 185: I would remove that it was recommended and just indicate that you used it, with evidence suggesting that mid-P provides…

Otherwise, great description of the tests you ran!

Results:

Lines 198-199: I see you included the DNA concentration here – this may be preference, I prefer it in the methods since it affects your results, but this may be an author preference (if it is not a journal preference). I still think it would be useful to include the details on those that were not able to be standardized. So, were 30 samples between 8.01 and 25ng/ul or were 300 samples in that range?

Line 204: detail “most closely” – was it 100% match? Not quite?

Line 120 Table: I see a stray “213” showing up in the background of the table

Lines 219-220: Might want to explain this in terms of your results a bit more to remind the reader what it means for your work rather than the data’s “discordant cells” just to help the reader put the results in perspective and interpret them a bit better.

Discussion:

Line 250: Is this result found in other viruses? Has the LTR gene been found to be more conservative or is it known to be more conservative? And if so, why might this be the case? These are great findings! Backing it up with additional evidence and reasoning will be helpful.

Line 236: Indicate “geographic” regions, rather than just regions because I first thought of gene region.

General: Might want to discuss the lack of standardization of some samples and how this could have played a role (though I’m not positive it would result in a difference between being positive for LTR vs gag, which is what you’re looking at, so it may not be necessary – I’m just thinking if it tested negative for gag but positive for LTR could it have something to do with concentration? But it was the same sample with the same concentration so likely not now that I think of it. Anyway, you may have some thoughts on this worth including).

6. PLOS authors have the option to publish the peer review history of their article (what does this mean?). If published, this will include your full peer review and any attached files.

Reviewer #1: **Yes: **Dustin Edwards

Reviewer #2: No

---

## [Author Response · Author response to Decision Letter 0]

10 Dec 2023

A word document containing responses to reviewer and editor comments is included in this submission. Below are the contents of said document.

Dear Dr. Pilotti,

Thank you for the opportunity to revise this paper. We appreciate the comments made by you and the reviewers. We have done our best to address the comments and incorporate the suggestions. Our responses to each comment provided in the decision letter are in indented text below. 

Academic Editor Comments: 

We reviewed the manuscript to ensure it and file names meet PLOS ONE’s style requirements.

Please review your reference list to ensure that it is complete and correct.

 The reference list was reviewed for completion and correctness. Additional references were added to support justification for the study

1. Abstract

Line 30: I would like to recommend removing "the portion" of the gag gene (or if you prefer to mention it, it would be better to specify the sequence) and replacing "virus' genome" with provirus if it's what the authors were referring to. Please consider following this suggestion in other sentences of the paper.

 We added specificity regarding the portion of the gag gene we targeted as “a 413-nucleotide sequence (partial p31/ partial capsid)”. We removed all occurrences of “the portion”. We replaced “virus’ genome” with provirus throughout the manuscript. Similarly, we added specificity to the LTR gene as “a 335-nucleotide sequence starting within in the U3 region of the LTR and extending into part of the Matrix of the gag region”.

Line 31: .......other tissues.... Could you specify?

 We included the other types of tissue that have been tested.

Line 32:..... different sections....Could you articulate better?

 We changed “different sections” to “different genomic regions” to clarify meaning.

Line 33: Could the authors please rephrase the purpose? I suggest changing the following part " of another section of the viral genome a portion of the" at most.

 We rephrased the text to increase clarity.

Line 69: I think the proviral genome definition is not necessary.

 We removed the definition.

Line 80: The authors should remove (LTRs)

 We removed “(LTR)” from the line.

Line 115: "Some sample...". Could you please clarify? A few samples? Do these poor-quality samples can lead to biased estimations?

 The percentage of samples left undiluted was included in the line. In the results we added that low concentrations did not appear to impact detection, as both genes were able to detect positives on either end of the range of DNA concentrations.

2. Methods.

Line 122: I suggest to explain what "slight modifications" are.

 We clarified that the modifications were to the PCR mix and reaction conditions, which were described in the following line.

Line 122-124: the authors should indicate the concentration instead of volume.

 Concentration of PCR reaction ingredients was included where appropriate.

Line 130: "Positive controls ......were confirmed LPDV positive." Could the authors mention the methodology? I would remove "undefined", which does not add significant experimental value.

 We made the suggested edit and specified how LPDV positive control samples were confirmed.

Line136: Could you please mention the gene the forward primer is complementary to?

 We included the forward and reverse sequences.

Line 142-143: Please, indicate the 3’ and 5’ ends of primers.

 The 3’ and 5’ primer ends were indicated in the text.

Line 149: The title of this section seems very general to me. You could divide the content into two separate sections: sensitivity and sensibility analysis, and statistical analysis.

 We divided analyses into the two suggested sections.

3. Results:

I suggest the authors expand the paragraph emphasizing the robustness of the results.

 We added more detail emphasizing the robustness of our results by increasing the description of our ability to detect LPDV across a range of DNA concentrations. We also provided more specificity with respect to the match of our sequencing data to the LPDV entries in the NCBI database. We further described how we interpret the results of our statistical analyses such that more samples were detected to have LPDV with LTR than with gag.

Line 199: "Infection status ...testing positive". Could you please explain which gene the samples are positive for?

 We rephrased the line so it was clear which genes were being discussed.

4. Discussion

Line 238: I suggest removing "it was not a perfect test" and leaving after the comma "it did not detect all cases".

 We edited the line as suggested.

As a general suggestion, I recommend adding more references whenever the authors mention literature data. I also suggest the authors avoid general terms and redundant definitions whenever possible.

 Additional references were added citing studies of LPDV in wild turkeys that were recently published as well as the addition of references related to variation in retrovirus detection depending on the genomic regions screened. We have worked to tighten up the language following your recommendations for terminology and have attempted to reduce redundancy.

Reviewer Comments to the Author

 Reviewer 1:

1. Line 197: has written that 924 samples were tested. However, on Line 201: 772 samples tested negative, and 153 tested positive, which adds up to 925 samples. Please confirm the sample size tested and results and clarify as needed.

Data were reviewed and the mistake was corrected: 924 was updated to 925.

2. Readers would benefit from seeing representative gels containing the resolved PCR products (positive and negative controls; positive and negative samples) as Supporting Information.

 We added a Supporting Information file that includes two representative gels – one for the gag fragment and one for the LTR fragment.

Reviewer 2:

Overall: This is a very interesting and thorough study. It is certainly enlightening that the LTR region was able to detect positives that were not detected by the standard gag gene. This is definitely useful to the field going forward. Nice work and well-written.

● I do think you could go into more detail in both the intro and the discussion about the viral genes, why the LTR region may be a more conserved region compared with the gag gene. Has this been shown in other viruses? I think further justification and explanation on this topic of the comparison of genes and viruses in general is required for the reader to appreciate the full interpretation and impact of this work.

We pursued the LTR region based on the recommendation of a colleague (Dr. Renshaw). He indicated that in his work with the gag region with samples of turkeys from both their eastern distribution (where much of the LPDV work was initially conducted) as well as turkeys from further west in the distribution that he was encountering false negatives using gag for samples from the western distribution and had identified fewer false negatives with the LTR/gag Matrix region suggesting that region is more conserved than the gag region for LPDV in turkeys. As such, we chose to evaluate that region’s utility for detecting LPDV in Iowa (Midwest USA). We cannot find any solid evidence in the literature to suggest that one region – overall – would be expected to be more conserved than another. So, really the choice was based on exploratory experiences by a colleague. We added a small amount of additional detail to the text further explaining this.

● Lastly, I might mention somewhere that you are detecting pro-viral DNA (you explain the incorporation of retroviruses into the genome well, just important to be clear that you are detecting the inserted DNA rather than the original RNA).

This is an excellent point. We tried to clarify our language in the text.

● Also, general comment, since you only use 924 samples, this should be your “n” rather than the 1,022 you started with.

The sample size was corrected as suggested.

Abstract:

Line 29: Could be worth also mentioning that blood and other tissues (organs, cloacal swabs) have also been used.

 We included the other types of tissue that have been tested.

 Line 31: Then here could mention that the viability of the various sample types mentioned above have been evaluated, but not the portion of the viral genome (enhancing your justification!).

 We edited the text to combine what was written with the provided suggestion.

Line 33: Use commas to separate the “a portion of the LTR” clause.

We revised the text as suggested to improve clarity.

Intro:

 Line 63: If being inclusive, also Shea et al. 2022 in JWD - https://doi.org/10.7589/JWD-D-21-00152 - 699 asymptomatic turkeys

 We included the provided article where indicated as well as other appropriate locations.

 Line 64: Could be useful to mention some nonspecific symptoms, and that Allison et al. 2014 did report that it can cause mortality just to further justify your work

 We provided some examples of non-specific symptoms and included that there has been some related mortality.

 Paragraph Lines 75-84: I think stronger justification is needed as to how the genes may differ in their detection ability. I’m not sure personal communication is a strong enough justification. Has it been shown that detection varies between genes in other viruses? I’m not disagreeing, but I do think the justification and evidence for the justification is slightly lacking here. The last line is informative and does help with the justification, but since your whole premise is assessing another gene, providing sound justification for this is important.

Additional references were added to support that there can be variation in detection rates between genes in other retroviruses. Please see our response above about how we came to choose LTR. We chose to pursue this LTR/matrix fragment because it seemed to be working better for a colleague at detecting LPDV, especially in turkeys further west. We added some text referring to another paper that has recently been published that used the env gene rather than the gag gene – to highlight that there are other groups examining other genomic regions besides the gag gene.

Line 85: It seems like you are referencing a diagnostic paper to indicate that wild turkey populations are declining. The reports of decline are somewhat isolated, and I would be more specific here and cite the sources that identified these declines.

This is a good point. Unfortunately, at this time, most of the reports of a decline are from natural resources biologist managers in the form of personal communications (i.e., in our case our Iowa Department of Natural Resources turkey biologist) or popular press articles (e.g., New York Times article “Turkeys Were a Marvel of Conservation. Now Their Numbers Are Dwindling.” Nov. 23, 2023) rather than rigorous analysis of long-term datasets or current studies examining population trends. Biologists are conducting such studies right now (including in Iowa). We toned down our language here.

 Methods:

 Line 106: Since you indicate below in line 112 that some samples had less initial material, it would be helpful to indicate the range of bone marrow mg used. Or you can add this into line 112 and put it in parenthesis, for instance: “If a small quantity of bone marrow was extracted (10-24mg), samples were instead eluted with 50ul (to increase yield).” Since you are specifically looking at detection probability, these somewhat specific details are important.

 We did not specifically measure the amount of bone marrow extracted partially out of concern for possible cross contamination among samples. The decision to elute using 50ul rather than 100ul was based off of relative quantities, comparing to the amount of bone marrow we typically used (i.e. if a sample had roughly half the amount of bone marrow that we normally used, or less, it was eluted with 50ul). Our primary concern was in decisions about creation of working stocks (standardization of DNA concentrations) which were based on DNA yield from quantification results because yield can be influenced by amount of starting material and also the quality of that material.

Lines 115-117: Again, useful to include the range of those samples. Were you potentially working with 1ng/ul? This may be a discussion point too if samples that had lower concentrations varied in their detection rates among genes.

 The range of concentrations for samples that were left undiluted was included.

 Lines 122-124: for primers and DNA, it is useful to use concentrations. You’ve already touched upon DNA concentrations, so at 50ng/ul you are using 100ng, but you did indicate this varies, so could include the range (though still note you attempted to standardize because this is good to know). Along these lines, it could be helpful to know how many were standardized, was it 90% of your samples? 10% of your samples? Providing these details help give the bigger picture of the work. For primers, it could be helpful to put the final concentration in parenthesis, or that you used 1ul of each 10uM primer.. Etc.

We included concentrations for primers and DNA, as well as the percentage of samples that were standardized.

 Line 135: Just curious why the 386 bp instead of the 413 typically seen with the gag gene and these primers?

 Good catch. This is an error. We apologize and we corrected it.

Lines 127-137: Could probably save some text here by indicating your negative control was water instead of DNA template, and then I’m not sure you have to indicate where you got the positive control, but just that it was confirmed positive via sequencing – you indicated confirmed positive, I assume that’s by sequencing? I might just add that tidbit.

 We shortened text regarding the negative control and removed text discussing the source of positive controls and added that they were confirmed positive through sequencing.

 Line 137: Could include the amount that you confirmed through sequencing (% of positive samples confirmed via sequencing).

 We included the subset of gag positive samples that were confirmed through sequencing.

 Line 140: Instead of processing, could be more specific and say the thermocycle protocol.

 We made the recommended edits.

 Lines 142-145: Depending on the journal it may be necessary to just indicate the 3’ and 5’ ends of primers. Also, did Renshaw develop these primers? I’m not sure if the journal may want some information as to the development of these primers at all.

 5’ and 3’ ends were indicated. Yes, these primers were developed by Renshaw and then we optimized the PCR protocol for them in our lab.

Line 146: Same comment as above in regards to providing the amount in the subset.

 We provided the number of samples that tested positive using LTR that were sequenced.

 Analysis:

 Line 185: I would remove that it was recommended and just indicate that you used it, with evidence suggesting that mid-P provides…

 Otherwise, great description of the tests you ran!

 We made the suggested edits.

 Results:

 Lines 198-199: I see you included the DNA concentration here – this may be preference, I prefer it in the methods since it affects your results, but this may be an author preference (if it is not a journal preference). I still think it would be useful to include the details on those that were not able to be standardized. So, were 30 samples between 8.01 and 25ng/ul or were 300 samples in that range?

 We left the DNA concentration in results, but also included the percentage of samples that were left unstandardized as well as the range of unstandardized concentrations to the methods.

 Line 204: detail “most closely” – was it 100% match? Not quite?

 We included the pairwise identity and percent identical range for Iowa strains with the closest match in GenBank.

 Line 120 Table: I see a stray “213” showing up in the background of the table

 We were not able to find the stray number, so we’re not certain how to correct this.

 Lines 219-220: Might want to explain this in terms of your results a bit more to remind the reader what it means for your work rather than the data’s “discordant cells” just to help the reader put the results in perspective and interpret them a bit better.

 We revised text to clarify what the results said about our data.

 Discussion:

 Line 250: Is this result found in other viruses? Has the LTR gene been found to be more conservative or is it known to be more conservative? And if so, why might this be the case? These are great findings! Backing it up with additional evidence and reasoning will be helpful.

 Please see comments above about why we chose to pursue the LTR region. 

Line 236: Indicate “geographic” regions, rather than just regions because I first thought of gene region.

We revised the text as suggested to improve clarity.

 General: Might want to discuss the lack of standardization of some samples and how this could have played a role (though I’m not positive it would result in a difference between being positive for LTR vs gag, which is what you’re looking at, so it may not be necessary – I’m just thinking if it tested negative for gag but positive for LTR could it have something to do with concentration? But it was the same sample with the same concentration so likely not now that I think of it. Anyway, you may have some thoughts on this worth including).

 We added some information that we hope clears up this concern. We had samples that tested positive at both the lower and upper ends of the DNA concentration range for both gag and LTR, otherwise this definitely would have been a concern we would have needed to consider with respect to the false negatives for gag and/or LTR.

---

## [Editor Report · Decision Letter 1]

20 Dec 2023

Detection of lymphoproliferative disease virus in Iowa Wild Turkeys (Meleagris gallopavo): Comparison of two sections of the proviral genome

PONE-D-23-30258R1

Dear Dr. Smith,

We’re pleased to inform you that your manuscript has been judged scientifically suitable for publication and will be formally accepted for publication once it meets all outstanding technical requirements.

Kind regards,

Elisabetta Pilotti

Academic Editor

PLOS ONE
---

## [Editor Report · Acceptance letter]

3 Feb 2024

PONE-D-23-30258R1 

PLOS ONE

Dear Dr. Smith, 

I'm pleased to inform you that your manuscript has been deemed suitable for publication in PLOS ONE. Congratulations! Your manuscript is now being handed over to our production team.

Kind regards, 

on behalf of

Dr. Elisabetta Pilotti 

Academic Editor

PLOS ONE